# Successive Short- and Long-Range Magnetic Ordering in Ba_2_Mn_3_(SeO_3_)_6_ with Honeycomb Layers of Mn^3+^ Ions Alternating with Triangular Layers of Mn^2+^ Ions

**DOI:** 10.3390/ma16072685

**Published:** 2023-03-28

**Authors:** Artem Moskin, Ekaterina Kozlyakova, Seung Hwan Chung, Hyun-Joo Koo, Myung-Hwan Whangbo, Alexander Vasiliev

**Affiliations:** 1Department of Low Temperature Physics and Superconductivity, Lomonosov Moscow State University, Moscow 119991, Russia; 2Functional Quantum Materials Laboratory, National University of Science and Technology “MISIS”, Moscow 119049, Russia; 3Department of Chemistry and Research Institute for Basic Sciences, Kyung Hee University, Seoul 02447, Republic of Korea; 4Department of Chemistry, North Carolina State University, Raleigh, NC 27695-8204, USA

**Keywords:** honeycomb lattice, triangular lattice, long-range magnetic order, short-range magnetic order, first principles’ calculations

## Abstract

Mixed-valent Ba_2_Mn^2+^Mn_2_^3+^(SeO_3_)_6_ crystallizes in a monoclinic *P2*_1_*/c* structure and has honeycomb layers of Mn^3+^ ions alternating with triangular layers of Mn^2+^ ions. We established the key parameters governing its magnetic structure by magnetization *M* and specific heat *C*_p_ measurements. The title compound exhibits a close succession of a short-range correlation order at *T_corr_* = 10.1 ± 0.1 K and a long-range Néel order at *T_N_* = 5.7 ± 0.1 K, and exhibits a metamagnetic phase transition at *T < T_N_* with hysteresis most pronounced at low temperatures. The causes for these observations were found using the spin exchange parameters evaluated by density functional theory calculations. The title compound represents a unique case in which uniform chains of integer spin Mn^3+^ (*S* = 2) ions interact with those of half-integer spin Mn^2+^ (*S* = 5/2) ions.

## 1. Introduction

Compounds of transition metal magnetic ions exhibit a wide range of phenomena, which are commonly grouped in terms of their spin values: quantum magnetism for systems of small spins and classical magnetism for systems of large spins. In both groups, the ground state that a magnetic compound reaches at low temperatures is governed largely by the dimensionality of its spin exchange interactions and also by the degree of geometrical spin frustration (hereafter, spin frustration) [1]. It quickly becomes complicated to analyze the properties of a magnetic system with magnetic ions of two (or more) different oxidation states and two (or more) different values of spins in terms of these two factors unless the relative values of its various spin exchanges are known. These days, it has become almost routine to determine the values of spin exchanges for any complex magnetic system by performing energy mapping analysis [2,3,4]. This method employs a number of broken-symmetry spin states of a given magnetic system, then evaluates their energies using a spin Hamiltonian made up of the spin exchanges, in addition using density functional theory (DFT) calculations and finally maps the relative energies of the broken-symmetry states from the spin Hamiltonian to those of the DFT calculations. In other words, this method relates the energy spectrum of a model Hamiltonian to that of an electronic Hamiltonian using a set of broken-symmetry states. Over the years, the energy mapping analysis has led to correct spin lattice models with which to understand the properties of numerous magnetic materials.

Among low-dimensional mixed-valent compounds of small spin, LiCu_2_O_2_ attracts attention as a type-II multiferroic since the formation of the long-range spiral magnetic order is accompanied by the emergence of ferroelectricity [5,6]. Another low-spin mixed-valent compound is NaV_2_O_5_ which undergoes a charge ordering transition, which gives rise to a spin gap in the magnetic excitation spectrum [7]. Among low-dimensional mixed-valent compounds of large spin, manganese-based systems are most extensively studied. Two magnetic orderings and a spin–flop transition were observed in the mixed-valent compound Mn_3_O(SeO_3_)_3_, the magnetic ion arrangement of which shows the intersection of octa-kagomé spin sublattices and staircase-kagomé spin sublattices [8]. The hollandite-type compound, Ba_1.2_Mn_8_O_16_, undergoes a magnetic transition at 40 K, which is significantly lower than the Weiss temperature of −385 K, a characteristic feature of high spin frustration. Strong spin frustration usually results from triangular arrangements of magnetic ions with antiferromagnetic spin exchange for each edge and/or competing ferromagnetic and antiferromagnetic interactions [9].

In the present work, we examine how to understand the magnetic properties of the mixed-valent manganese compound Ba_2_Mn^2+^Mn^3+^_2_(SeO_3_)_6_ [10]. In general, spin exchanges between transition-metal magnetic ions M forming ML_n_ polyhedra with surrounding main-group ligands L are classified into M-L-M and M-L…L-M types [2,3,4], and the latter are further differentiated depending on whether or not the L…L bridge is coordinated by a metal cation A^m+^ to form L… A^m+^…L. In cases when a metal cation is present, the M-L…A^m+^…L-M exchanges are further differentiated by whether or not the A^m+^ cation is a d^0^ transition-metal cation or a main-group cation. The spin exchanges involving a main-group cation are almost impossible to predict using simple qualitative arguments, especially when this cation makes strong covalent bonds with the ligand to form a molecular anion, e.g., a P_2_S_6_^4−^ anion in MPS_3_ (M = Mn, Fe, Co, Ni) [11]. Such is the case for the SeO_3_^2−^ anion of Ba_2_Mn_3_(SeO_3_)_6_. As will be described below, the spin exchanges of Ba_2_Mn_3_(SeO_3_)_6_ are all of the Mn-O…Se^4+^…O-Mn type; that is, none involves the Mn-O-Mn type so that the Mn^3+^ and Mn^2+^ ions of Ba_2_Mn_3_(SeO_3_)_6_ do not generate double exchanges [12]. The Mn^2+^ ions of Ba_2_Mn_3_(SeO_3_)_6_ form chains, with the Mn^3+^ ions in the same direction. Between these chains, triangular arrangements of the magnetic ions such as (Mn^2+^, Mn^2+^, Mn^3+^) and (Mn^2+^, Mn^3+^, Mn^3+^) occur in all three directions, so one might consider the presence of strong spin frustration and exhibit magnetic properties expected for a low-dimensional antiferromagnetic material. To interpret these seemingly puzzling aspects of Ba_2_Mn_3_(SeO_3_)_6_, we determine its spin exchanges using the energy-mapping analysis to find the cause for its low-dimension antiferromagnetic behavior.

## 2. Materials and Methods

Mixed-valent Ba_2_Mn^2+^Mn_2_^3+^(SeO_3_)_6_ was synthesized by a hydrothermal reaction of BaCO_3_ (2 mmol), MnCl_2_∙4H_2_O (1 mmol) and H_2_SeO_3_ (3 mmol) as precursors with 1.5 mL of 65% HNO_3_ and 3 mL of water added. The mixture was placed into a Teflon chamber of a steel autoclave (10 mL) after the degassing was finished. The autoclave then was placed into the furnace, where the temperature was raised to 200 °C for 1 week. After this, the brown powder of Ba_2_Mn_3_(SeO_3_)_6_ was rinsed with water to wash out the contaminants. The obtained powder sample was found to crystallize in the monoclinic *P2*_1_*/c* space group with *a* = 5.4717(3) Å, *b* = 9.0636(4) Å, *c* = 17.6586(9) Å, *β* = 94.519(1), *V* = 873.03(8) Å^3^, *Z* = 2 in agreement with the original solution [10] and its powder XRD pattern (BRUKER D8 Advance diffractometer Cu Kα, λ = 1.54056, 1.54439 Å, LYNXEYE detector) is shown in Figure 1.

As depicted in Figure 2a, the structure is composed of MnO_6_ octahedra interlinked with SeO_3_ pyramids. The Mn1O_6_ octahedra of Mn^2+^ (*S* = 5/2) ions form trigonal layers, and Mn2O_6_ octahedra of Mn^3+^ (*S* = 2) ions form honeycomb layers, as shown in Figure 2b,c.

Physical properties of Ba_2_Mn_3_(SeO_3_)_6_ were characterized by measuring the magnetization *M* and the specific heat *C*_p_ on ceramic samples (well-pressed pellets of 3 mm in diameter and 1 mm in thickness) using various options of “Quantum Design” Physical Properties Measurements System PPMS—9 T taken in the range 2–300 K under magnetic field *µ*_0_*H* up to 9 T.

## 3. Results

### 3.1. Magnetic Susceptibility

The magnetic susceptibility *χ* = *M*/*H* of Ba_2_Mn_3_(SeO_3_)_6_ taken at *µ*_0_*H* = 0.1 T in the field-cooled regime is shown in Figure 3. In the high-temperature region, it follows the Curie–Weiss law:(1)$$\chi =\frac{C}{T-\theta }+\chi_{0}$$
with the temperature-independent term $\chi_{0}$ = −7.6 × 10^−4^ emu/mol, the Curie constant *C* = 10.98 emu K/mol and the Weiss temperature *Θ* = −27.8 K. The value of $\chi_{0}$ exceeds the sum of the Pascal constants of ions and groups constituting the title compound $\chi_{0,calc}$ = −3.5 × 10^−4^ emu/mol [13]. This should be attributed to the effect of sample holder. The value of *C* somewhat exceeds the value *C_calc_* = 10.375 emu K/mol expected under the assumption of g-factor, *g* = 2, for both the Mn^2+^ and Mn^3+^ ions. Use of *g* = 2 is reasonable for Mn^2+^ (*S* = 5/2) ions with no orbital-moment contribution, but it underestimates the *g*-factor for Mn^3+^ ions (*S* = 2). The negative value of the Weiss temperature *Θ* points to the predominance of antiferromagnetic exchange interactions at elevated temperatures. Its absolute value can be influenced by the competition of ferromagnetic and antiferromagnetic exchange interactions.

On lowering the temperature, the *χ(T)* curve passes through a broad maximum at *T_corr_* = 10.2 K and shows a kink at *T_N_* = 5.6 K, which is more pronounced in the Fisher’s specific heat *d(χT)/dT* (not shown). This broad maximum is typically found for a quasi-one-dimensional (1D) antiferromagnetic chain system; hence, suggesting that Ba_2_Mn_3_(SeO_3_)_6_ has a 1D-like antiferromagnetic subsystem. The kink at a lower temperature shows that Ba_2_Mn_3_(SeO_3_)_6_ undergoes a long-range antiferromagnetic order. The drop of magnetic susceptibility *χ* below its largest value at *T*_corr_ is less than one-third of that expected for a three-dimensional easy-axis antiferromagnet [14]. The absence of a so-called Curie tail at lowest temperatures signals the high chemical purity of the sample.

### 3.2. Field Dependence of Magnetization

The field dependencies of the magnetization *M* taken at selected temperatures in the *T < T_N_* and *T_N_ < T < T_corr_* regions are shown in Figure 4. At the highest temperature of our measurement, the *M(H)* curve is linear indicating that the system is in the paramagnetic state, but starts to deviate from linearity as the temperature is lowered toward *T_N_*. Below *T_N_*, the *M(H)* curves exhibit hysteresis, which becomes most pronounced at the lowest temperature of our measurement. In general, the Heisenberg magnets of quasi-isotropic magnetic moment experience a spin–flop transition prior to the full saturation at the spin–flip transition. This is not the case for Ba_2_Mn_3_(SeO_3_)_6_, although it has isotropic Mn^2+^ ions. Instead, it exhibits a metamagnetic transition inherent to the Ising magnets. Such behavior should be associated with the presence of highly anisotropic Mn^3+^ ions in the system. The well-pronounced hysteresis underlines the first-order nature of the metamagnetic transition [15].

### 3.3. Heat Capacity

The magnetization data are fully consistent with the specific heat data, shown in Figure 5. In a wide temperature range, the *C_p_(T)* curve can be described by the sum of the Debye function with *Θ_D_* = 223 K and two Einstein functions with *Θ_E_*_1_ = 556 K and *Θ_E_*_2_ = 1449 K. The first of the Einstein functions can be ascribed to the oscillation mode of the MnO_6_ octahedra and the second one to that of the SeO_3_ pyramids. These parameters were obtained by fitting the data in the 70–290 K region with the fixed sum of the Debye and Einstein functions. The remaining data were considered as a purely magnetic contribution. Indeed, the magnetic entropy is nearly equal to the theoretical value of *R*(2ln(5) + ln(6)) = 41.6 J/mol K, confirming the accuracy of the fit. Nevertheless, a nonmagnetic analogue is still needed to obtain more accurate values of Debye and Einstein temperatures.

On lowering the temperature, the specific heat *C_p_* passes through a broad maximum at *T_corr_* = 10 K and shows a peak at *T_N_* = 5.8 K. Under external magnetic field, the broad maximum retains its position, but the sharp anomaly shifts to lower temperatures. Such behavior is typical of low-dimensional antiferromagnets experiencing successive short-range and long-range orders.

### 3.4. Spin Exchanges and Interpretation

The two important issues concerning the observed magnetic properties of Ba_2_Mn_3_(SeO_3_)_6_ are the cause for the broad maximum of the magnetic susceptibility at *T_corr_* = 10.1 ± 0.1 K, suggesting a short-range correlation as found for a 1D antiferromagnetic chain and a sharp kink at 5.7 ± 0.1 K, suggesting a long-range antiferromagnetic ordering. With *Θ* = −27.8 K and *T_N_* = 5.7 K (the index of spin frustration *f* = 5.0), the spin frustration in Ba_2_Mn_3_(SeO_3_)_6_ is not strong enough to prevent it from adopting a long-range antiferromagnetic ordering. This is somewhat surprising because one might expect a strong spin frustration in Ba_2_Mn_3_(SeO_3_)_6._ Figure 2c shows that the interaction between a chain of Mn^2+^ ions with the Mn^3+^ ions in the surrounding hexagonal prism generates numerous spin exchange triangles, which is a common arrangement leading to spin-frustration. Furthermore, these interactions must give rise to a 1D antiferromagnetic chain behavior to explain the 1D-like short range correlation at 10.1 K. To explore these issues, we first evaluate the spin exchanges of various exchange paths *J* in Ba_2_Mn_3_(SeO_3_)_6_.

All adjacent magnetic ions of Ba_2_Mn_3_(SeO_3_)_6_ are interconnected by the SeO_3_ pyramids except for the Mn^3+^ ions encircled by dashed ellipses in Figure 2a. The O…O contact distances (3.988 Å) of their Mn-O…O-Mn exchange paths are well beyond the van der Waals distance of ~3.30 Å so these spin exchanges can be neglected. There are still numerous spin exchanges between the Mn^2+^ and Mn^3+^ ions as depicted in Figure 6a. The spin exchange paths *J*_2_ (*J*_1_) form chains of Mn^3+^ (Mn^2+^) ions along the *a*-direction (Figure 6b). For convenience, these chains will be referred to as *J*_2_- and *J*_1_- chains, respectively. Note that each *J*_2_-chain is coupled to two adjacent *J*_2_-chains and also to two adjacent *J*_1_-chains. Between adjacent *J*_1_- and *J*_2_-chains four different spin exchange paths (i.e., *J*_4_, *J*_5_, *J*_6_ and *J*_7_) occur (Figure 6c), leading to (*J*_1_, *J*_4_, *J*_5_), (*J*_2_, *J*_4_, *J*_5_), (*J*_1_, *J*_6_, *J*_7_) and (*J*_2_, *J*_6_, *J*_7_) exchange triangles. A more extended view of Figure 6c is presented in Appendix A in the Appendix A.

To determine the values of these exchanges, we employ the spin Hamiltonian defined as
(2)$$H_{spin}=-\sum_{i>j}J_{ij}{\vec{S}}_{i}\cdot {\vec{S}}_{j},$$
where the spin exchange *J_ij_* between two spin sites can be any one of *J*_1_*–J*_7_. To evaluate *J*_1_*–J*_7_, we carry out the energy-mapping analysis [2,3,4] using the eight ordered spin states, i.e., AFi, where *i* = 1–8, depicted in Appendix A. First, we express the energies of the eight ordered states in terms of the spin exchanges *J*_1_–*J*_7_ using the spin Hamiltonian of Equation (2) and then determine the relative energies of these states (Table 1) by DFT calculations using the frozen core projector augmented plane wave [16,17] encoded in the Vienna ab Initio Simulation Package [18] and the exchange-correlation functional of Perdew, Burke and Ernzerhof [19].

The electron correlations associated with the 3*d* states of Mn were taken into consideration by DFT + *U* calculations with effective on-site repulsion *U_eff_* = *U* − *J* = 3 eV and 4 eV [20]. All our DFT + U calculations used the plane wave cutoff energy of 450 eV, a set of (6 × 4 × 4) k-points, and the threshold of 10^−6^ eV for self-consistent-field energy convergence. Finally, the numerical values of *J*_1_–*J*_7_ were obtained by mapping the relative energies of the eight ordered spin states onto the corresponding energies determined by DFT + *U* calculations. The results of these energy-mapping analyses are summarized in Table 2.

As already pointed out, each *J*_2_-chain interacts with two adjacent *J*_2_-chains and with two adjacent *J*_1_-chains (Figure 6b). In terms of the spin exchanges of Table 2, the nature of these interchain interactions can be stated as follows:

(1) Each *J*_1_-chain is an antiferromagnetic chain, and so is each *J*_2_-chain.

(2) Each *J*_2_-chain is ferromagnetically coupled to two adjacent *J*_2_-chains via the exchange *J*_3_ (Figure 7a).

(3) Each *J*_2_-chain is coupled to one *J*_1_-chain via the antiferromagnetic exchange *J*_6_ and the ferromagnetic exchange *J*_7_ (Figure 7b), forming the (*J*_1_, *J*_6_, *J*_7_) and (*J*_2_, *J*_6_, *J*_7_) exchange triangles. With one ferromagnetic and two antiferromagnetic exchanges, each exchange triangle is not spin-frustrated, so the coupling between these *J*_2_- and *J*_1_-chains is ferromagnetic.

(4) Each *J*_2_-chain is coupled to another *J*_1_-chain via the antiferromagnetic exchanges *J*_4_ and *J*_5_ (Figure 7c), forming the (*J*_1_, *J*_4_, *J*_5_) and (*J*_2_, *J*_4_, *J*_5_) exchange triangles. With all three antiferromagnetic spin exchanges, each exchange triangle is spin frustrated. Thus, as depicted in Figure 7c,d, one can have two different spin arrangements between these *J*_1_- and *J*_2_-chains. This explains the presence of spin frustration in Ba_2_Mn_3_(SeO_3_)_6_ as indicated by its index of spin frustration of *f* = 5. Since *J*_4_ is more strongly antiferromagnetic than *J*_5_ (by a factor of approximately 3), the spin configuration of Figure 7c is energetically more stable than that of Figure 7d. The antiferromagnetic ordering at *T_N_* = 5.7 K means that the spin configuration of Figure 7c dominates over that of Figure 7d in the population.

(5) As already noted, each *J*_2_-chain is an antiferromagnetic chain and interacts with two adjacent *J*_2_-chains and two adjacent *J*_1_-chains. These interchain interactions are all ferromagnetic except for the one with one of the two *J*_1_-chains. The latter is spin-frustrated as described above. Above *T_N_* = 5.7 K, where the latter spin frustration is not settled, the magnetic behavior of Ba_2_Mn_3_(SeO_3_)_6_ should have a strong 1D antiferromagnetic chain character because the antiferromagnetic *J*_1_- and *J*_2_-chains are ferromagnetically coupled (via ferromagnetic *J*_7_ and antiferromagnetic *J*_6_, Figure 6b). This explains the occurrence of the broad maximum in the magnetic susceptibility and the specific heat of Ba_2_Mn_3_(SeO_3_)_6_. However, interactions between the *J*_1_- and *J*_2_-chains via *J*_4_ and *J*_5_ are spin-frustrated, because the (*J*_1_, *J*_4_, *J*_5_) and (*J*_2_, *J*_4_, *J*_5_) exchange triangles are spin-frustrated so two different arrangements between the *J*_1_- and *J*_2_-chains are possible (Figure 6c,d).

The two sets of the spin exchanges obtained with *U*_eff_ = 3 and 4 eV are similar in trend. To see which set is better, one might estimate the Weiss temperature *Θ* using the mean field theory [21,22], to see which set leads to a value closer to the experimental value of *Θ* = −27.8 K observed for Ba_2_Mn_3_(SeO_3_)_6_. According to Figure 6a,c, the *Θ*_*calc*_ value for Mn^2+^ (*S* = 5/2) ions is predicted to be
(3)$$\Theta_{calc}=\frac{S\left(S+1\right)}{3}(2J_{1}+2J_{4}+2J_{5}+2J_{6}+2J_{7}),$$
which is –21.9 K for *U*_eff_ = 3 eV and –9.9 K for *U*_eff_ = 4 eV. Similarly, the *Θ*_*calc*_ value for Mn^3+^ (*S* = 2) ions is predicted to be
(4)$$\Theta_{calc}=\frac{S\left(S+1\right)}{3}(2J_{2}+2J_{3}+2J_{4}+2J_{5}+2J_{6}+2J_{7})$$
which is −10.2 K for *U*_eff_ = 3 eV and −1.2 K for *U*_eff_ = 4 eV. Thus, the spin exchanges obtained from of *U*_eff_ = 3 eV better describes the observed Weiss temperature.

## 4. Discussion and Conclusions

In summary, our magnetization and specific heat measurements of Ba_2_Mn_3_(SeO_3_)_6_ reveal that it is a low-dimensional antiferromagnet with a short-range one-dimensional antiferromagnetic chain behavior followed by a long-range antiferromagnetic order as marked by a succession of a broad maximum at *T*_corr_ = 10.1 ± 0.1 K and a sharper anomaly at *T*_N_ = 5.7 ± 0.1 K in both magnetic susceptibility (Fisher’s specific heat) and specific heat. These observations are well-explained in terms of the spin exchanges of Ba_2_Mn_3_(SeO_3_)_6_, both antiferromagnetic and ferromagnetic, evaluated by the energy-mapping analysis. In the ordered state, Ba_2_Mn_3_(SeO_3_)_6_ exhibits a metamagnetic phase transition inherent for the Ising magnets with magnetization hysteresis most pronounced at low temperatures. Notably, no hysteresis is observed at *µ*_0_*H* = 0 which points to the absence of spontaneous magnetization in Ba_2_Mn_3_(SeO_3_)_6_.

Structurally, the title compound is organized by the honeycomb layers of Mn^3+^ ions alternating with the triangular layers of Mn^2+^ ions. Magnetically, it consists of uniform chains of integer spins *S* = 2 of Mn^3+^ ions and half-integer spins *S* = 5/2 of Mn^2+^ ions running along the *a* axis. Qualitatively different quantum ground states, i.e., gapped and gapless spin liquid, correspondingly, can be expected for these entities [23]. However, the interaction of these chains leads to the formation of a long-range antiferromagnetic order. It would be interesting to synthesize and study isostructural phases of Ba_2_Mn_3_(SeO_3_)_6_ where either the divalent or the trivalent magnetic ions is replaced with nonmagnetic counterparts. The exchange interactions through a chalcogenide anion such as SeO_3_^2-^ makes the scales of magnetic fields and temperatures quite convenient for experiments with equipment readily available.

While preparing this article, we became aware of an independent unpublished study on Ba_2_Mn_3_(SeO_3_)_6_ [24], which reported experimental data similar to ours but did not provide any theoretical analysis.

## Figures and Tables

**Figure 1 materials-16-02685-f001:**
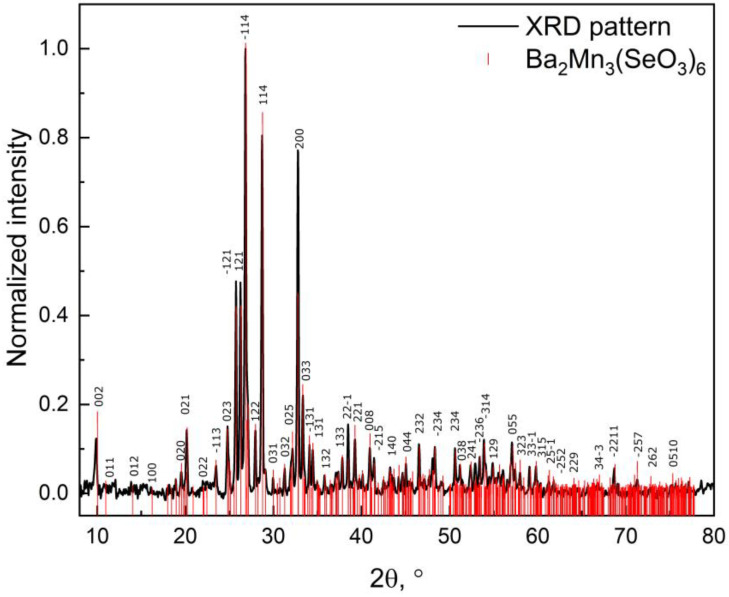
Diffraction pattern obtained for powder samples of Ba_2_Mn_3_(SeO_3_)_6_. The red lines are shown as the reference for peak positions and indexing (some of the indexes were omitted due to high density of peaks).

**Figure 2 materials-16-02685-f002:**
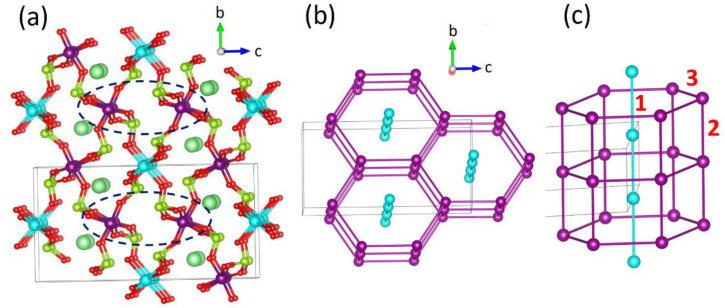
(**a**) The crystal structure of Ba_2_Mn_3_(SeO_3_)_6_, where the Mn1 and Mn2 atoms are represented by cyan and violet spheres, respectively, and the Ba and Se atoms by large and small green spheres, respectively. The Mn2O_6_ octahedra encircled by dashed ellipses have no interlinking by a SeO_3_ pyramid. (**b**) The arrangements of the Mn1 and Mn2 atoms in Ba_2_Mn_3_(SeO_3_)_6_. (**c**) One hexagonal prism of Mn^3+^ ions containing one chain of Mn^2+^ ions. The number 1 refers to the spin exchange *J*_1_ of the chain of Mn^2+^ ions, while 2 and 3 refer, respectively, to spin exchanges *J*_2_ and *J*_3_ of the hexagonal prism of Mn^3+^ ions (see below).

**Figure 3 materials-16-02685-f003:**
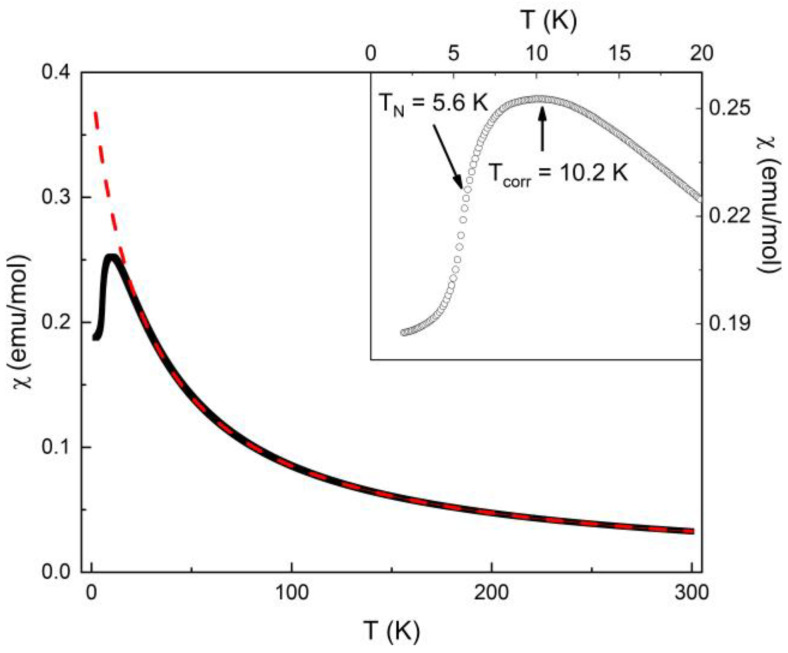
Temperature-dependent magnetic susceptibility *χ* of Ba_2_Mn_3_(SeO_3_)_6_ taken at *µ*_0_*H* = 0.1 T in the field-cooled regime. The dashed line represents the extrapolation of the Curie–Weiss fit. The inset shows a zoomed-in view of the low temperature region of the *χ*(*T*) curve.

**Figure 4 materials-16-02685-f004:**
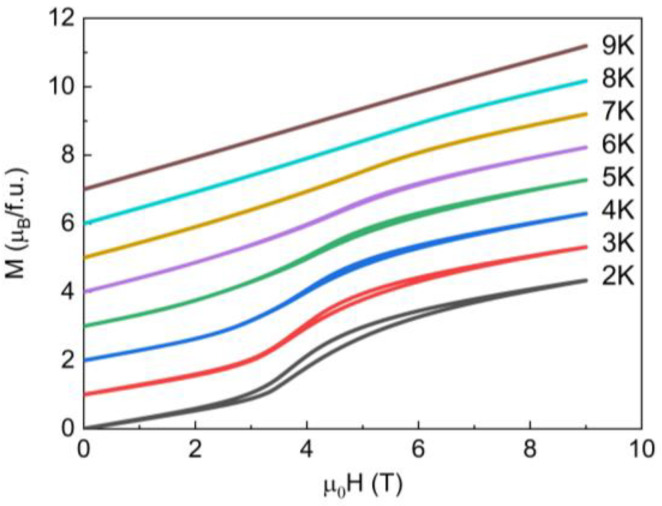
Magnetization curves measured for Ba_2_Mn_3_(SeO_3_)_6_ at selected temperature in the *T < T_N_* and *T_N_ < T < T_corr_* regions. For clarity, the curves are shifted with respect to each other by 1 *µ*_B_/f.u.

**Figure 5 materials-16-02685-f005:**
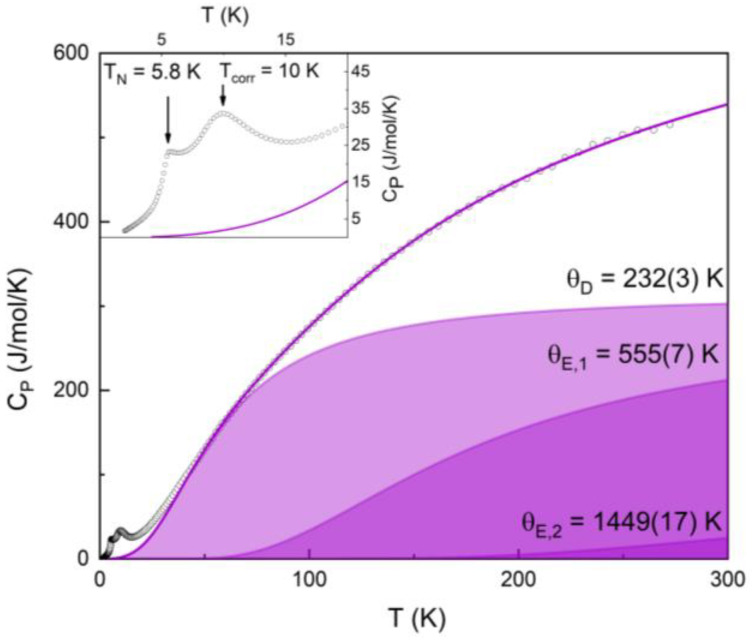
Temperature-dependent specific heat of Ba_2_Mn_3_(SeO_3_)_6_ described by the sum of Debye and two Einstein functions. The inset shows a zoomed-in view of the low temperature region along with the fitting curve.

**Figure 6 materials-16-02685-f006:**
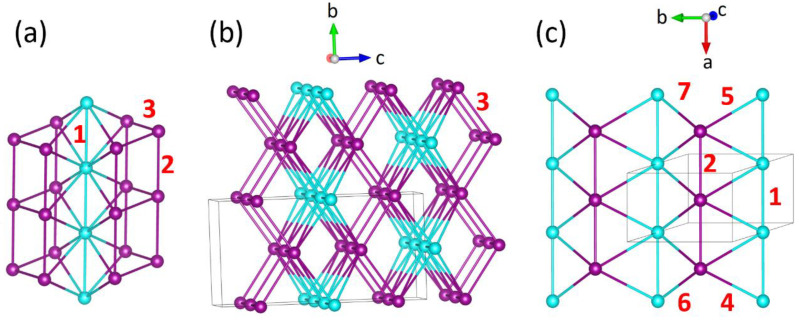
Arrangement of the Mn^2+^ (cyan) and Mn^3+^ (violet) ions in Ba_2_Mn_3_(SeO_3_)_6_, where the numbers 1–7 refer to the exchange paths *J*_1_–*J*_7_, respectively: (**a**) Spin exchange paths between the Mn^2+^/Mn^3+^ ions that are bridged by the SeO_3_ groups in one hexagonal prism of Mn^3+^ ions with one chain of Mn^2+^ ions. (**b**) Projection view showing how chains of Mn^3+^ ions interact with those of Mn^2+^ ions. (**c**) Spin exchange paths *J*_4_–*J*_7_ between the Mn^2+^ and Mn^3+^ ions.

**Figure 7 materials-16-02685-f007:**
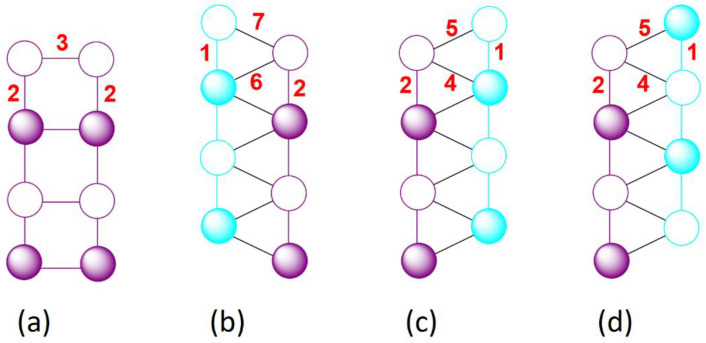
(**a**) Ferromagnetic coupling between two adjacent *J*_2_-chains. (**b**) Ferromagnetic coupling between two adjacent *J*_1_- and *J*_2_-chains when the (*J*_1_, *J*_6_, *J*_7_) and (*J*_2_, *J*_6_, *J*_7_) exchange triangles are not spin-frustrated. (**c**,**d**) Two possible spin arrangements between two adjacent *J*_2_- and *J*_1_-chains when the (*J*_1_, *J*_4_, *J*_5_) and (*J*_2_, *J*_4_, *J*_5_) exchange triangles are spin-frustrated.

**Table 1 materials-16-02685-t001:** Relative energies (in meV/FU) of the eight ordered spin states obtained from DFT + *U* calculations.

Ordered Spin States	*U_eff_* = 3 eV	*U_eff_* = 4 eV
AF_1_	10.01	8.27
AF_2_	8.80	7.20
AF_3_	6.40	5.32
AF_4_	8.49	7.81
AF_5_	17.84	15.03
AF_6_	13.26	12.68
AF_7_	3.62	3.29
AF_8_	0	0

**Table 2 materials-16-02685-t002:** Geometrical parameters of the exchange paths and the values of the spin exchanges in Ba_2_Mn_3_(SeO_3_)_6_. The plus and minus signs of the spin exchanges represent ferromagnetic and antiferromagnetic couplings, respectively.

Path	Geometrical Parameters		Spin Exchanges (in K)	
	Ions Involved	Distance, Å	*U_eff_* = 3 eV	*U_eff_* = 4 eV
*J* _1_	Mn1…Mn1	5.4717	−2.23	−1.74
*J* _2_	Mn2…Mn2	5.4717	−2.62	−2.14
*J* _3_	Mn2…Mn2	5.4655	1.51	1.80
*J* _4_	Mn1…Mn2	5.9554	−2.38	−1.63
*J* _5_	Mn1…Mn2	6.0883	−0.88	−0.45
*J* _6_	Mn1…Mn2	5.9737	−2.05	−1.43
*J* _7_	Mn1…Mn2	6.1063	3.88	3.56

## Data Availability

The data are available on reasonable request.

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
