# Peer review of "Successive Short- and Long-Range Magnetic Ordering in Ba2Mn3(SeO3)6 with Honeycomb Layers of Mn3+ Ions Alternating with Triangular Layers of Mn2+ Ions"

_materials, 2023, doi:10.3390/ma16072685_

Round 1
Reviewer 1 Report
The work describes a proposed Mn honeycomb structure and interesting properties are possible from the chain-like arrangements. But I need to see more evidence for this structure.
1. Structural characterization is missing. The most critical one is XRD, and the data were not shown, only the lattice parameters. Please show the XRD pattern and index the peaks.
2. The low dimensional, short range AFM feature was assigned to 1D chains. Is it possible coming from ordering on the 2D lattice or other 2D features? For example, in-plane ordering can form before the interplane ordering.
3. It feels to me the exchange 4567 are same length, angle, atoms. Why 7 is so different?
Overall this is a good work and could be published.
Author Response
Details of revision and Replies
In our revision, the comments of the three reviewers were accommodated as described below. We thank their comments, which helped improve our work.
[1] Reviewer 1
The work describes a proposed Mn honeycomb structure and interesting properties are possible from the chain-like arrangements. But I need to see more evidence for this structure.
Comment 1:
Structural characterization is missing. The most critical one is XRD, and the data were not shown, only the lattice parameters. Please show the XRD pattern and index the peaks.
Reply 1:
The XRD pattern was included as Figure 1 with indexed peaks.
Comment 2:
The low dimensional, short range AFM feature was assigned to 1D chains. Is it possible coming from ordering on the 2D lattice or other 2D features? For example, in-plane ordering can form before the interplane ordering.
Reply 2:
Each chain is AFM-coupled. It is the interaction between chain that is spin-frustrated. Thus, it is natural to discuss in terms of chains.
Comment 3:
It feels to me the exchange 4567 are same length, angle, atoms. Why 7 is so different?
Reply 3:
The distances are listed to help identify the spin exchange paths examined. Spin exchanges are strongly influenced by the overlap between magnetic orbitals, which may be unrelated to the metal…metal distances.
Overall this is a good work and could be published.
Reviewer 2 Report
The authors study the compounds Ba2Mn3(SeO3)6, which presents low-dimensional chain magnetism and spin triangles. They perform synthesis, experimental structural and magnetism study, and mapping of first-principles calculations of relative energies to exchange constants in a spin Hamiltonian. Unfortunately some of the same measurements have been done by another group, as noted by the authors at the end of this manuscript, which reduces the novelty. Nevertheless I recommend its publication, after minor corrections and suggestions are considered.
Text
Line 86 “Mn3p” is it 3+?
“precursors+ with”, “+ ions Teflon” I think “+” is a typo here
116 χ0
118 χ0calc
0 and 0calc should be subscripts
95 Suggestion: It is probably less confusing to call the different Mn directly Mn2+ and Mn3+, instead of using also the numbered Mn1, Mn2
The equations are badly formatted. Low image quality (pixelated) and equation number (1) is missing.
136 “The drop of magnetic susceptibility χ below its largest value at Tcorr is less than one-third expected for a three-dimensional easy-axis antiferromagnet.”
Is there a reference for the expected value?
217 "Packages" is "Package".
There is text overlapping with table 2 ("As already"). This table is also incorrectly numbered table 1.
164 R(2ln(5) + ln(6)) 41.6 J/mol K
Missing equal sign.
180 "and a sharp maximum at 5.7 ± 0.1 K" There is no maximum in the susceptibility curve (Fig. 2) at T_N. I believe you mean Fisher’s specific heat, ∂(χT)/∂T
Figure 5 caption suggestion: I would repeat here which Mn corresponds to which color, for easier reference.
254 “Each J2-chain is coupled to another J1-chain via the antiferromagnetic exchanges J5 and J6 (Figure 6b), forming the (J1, J4, J5) and (J2, J4, J5) exchange triangles.”
It should be “via the antiferromagnetic exchanges J4 and J5“, to be consistent with the exchange triangles in the second part of the sentence, and Figure 6b does not show those interactions, it is 6c.
278 “To see if which set” should be “To see which”
Ref 12 author Zener
Suggestion: the high-spin orbital interactions article
H.-J. Koo, R. K. Kremer, and M.-H. Whangbo, High-Spin Orbital Interactions Across van Der Waals Gaps Controlling the Interlayer Ferromagnetism in van Der Waals Ferromagnets, J. Am. Chem. Soc. 144, 16272 (2022).
could be cited in the introduction.
Author Response
Details of revision and Replies
In our revision, the comments of the three reviewers were accommodated as described below. We thank their comments, which helped improve our work.
[2] Reviewer 2
The authors study the compounds Ba2Mn3(SeO3)6, which presents low-dimensional chain magnetism and spin triangles. They perform synthesis, experimental structural and magnetism study, and mapping of first-principles calculations of relative energies to exchange constants in a spin Hamiltonian. Unfortunately, some of the same measurements have been done by another group, as noted by the authors at the end of this manuscript, which reduces the novelty. Nevertheless, I recommend its publication, after minor corrections and suggestions are considered.
Comment 1:
Line 86 “Mn3p” is it 3+?
Reply:
The typos were corrected.
Comment 2:
“precursors+ with”, “+ ions Teflon” I think “+” is a typo here
Reply 2:
The typos were corrected
Comment 3
116 χ0
Reply 3:
The typo was corrected.
Comment 4:
118 χ0calc, 0 and 0calc should be subscripts
Reply 4:
The typo was corrected.
Comment 5:
95 Suggestion: It is probably less confusing to call the different Mn directly Mn2+ and Mn3+, instead of using also the numbered Mn1, Mn2
Reply 5:
In describing the crystal structure, we follow the convention of using Mn1 and Mn2. Whether these sites have Mn2+ or Mn3+ ions is another matter. Thus, we prefer to employ the conventional description.
Comment 6:
The equations are badly formatted. Low image quality (pixelated) and equation number (1) is missing.
Reply 6:
We do not have this issue. Maybe a problem with Word version?
Comment 7:
136 “The drop of magnetic susceptibility χ below its largest value at Tcorr is less than one-third expected for a three-dimensional easy-axis antiferromagnet.”
Is there a reference for the expected value?
Reply 7:
- Blundell, Oxford University Press, 2001, 272 pp.
Comment 8:
217 "Packages" is "Package".
Reply 8:
The typo was corrected.
Comment 9:
There is text overlapping with table 2 ("As already"). This table is also incorrectly numbered table 1.
Reply 9:
The typos were corrected.
Comment 10:
164 R(2ln(5) + ln(6)) 41.6 J/mol K
Missing equal sign.
Reply 10:
The typo was corrected.
Comment 11:
180 "and a sharp maximum at 5.7 ± 0.1 K" There is no maximum in the susceptibility curve (Fig. 2) at T_N. I believe you mean Fisher’s specific heat, ∂(χT)/∂T.
Reply 11:
‘maximum’ was replaced with ‘kink’.
Comment 12:
Figure 5 caption suggestion: I would repeat here which Mn corresponds to which color, for easier reference.
Reply 12:
That information was added.
Comment 13:
254 “Each J2-chain is coupled to another J1-chain via the antiferromagnetic exchanges J5 and J6 (Figure 6b), forming the (J1, J4, J5) and (J2, J4, J5) exchange triangles.”
It should be “via the antiferromagnetic exchanges J4 and J5“, to be consistent with the exchange triangles in the second part of the sentence, and Figure 6b does not show those interactions, it is 6c.
Reply 13:
That part was rewritten to clarify the difference of Figure 6b from Figure 6c, d.
Comment 14:
278 “To see if which set” should be “To see which”
Reply 14:
The typo was corrected.
Comment 15:
Ref 12 author Zener
Reply 15:
The typo was corrected.
Comment 16:
Suggestion: the high-spin orbital interactions article
H.-J. Koo, R. K. Kremer, and M.-H. Whangbo, High-Spin Orbital Interactions Across van Der Waals Gaps Controlling the Interlayer Ferromagnetism in van Der Waals Ferromagnets, J. Am. Chem. Soc. 144, 16272 (2022). could be cited in the introduction.
Reply 16:
It was difficult to cite this reference because our work does not deal with van der Waals ferromagnets.
Reviewer 3 Report
The paper by Moskin et al. reports on the structural and magnetic properties of Ba2Mn3(SeO3)6. In particular the combination of experimental and theoretical analysis yields a deep insight into the short- and long-range magnetic coupling and spin exchange phenomena. On that basis the properties of this particular material can largely be understood. Furthermore, obtained results will be helpful to understand the properties of structurally similar or closely related materials.
A minor flow of the presentation is the mixing of cgs and SI units, which could definitely be avoided. Furthermore, the susceptibility=M/H is usually a dimensionless quantity rather than being measured in emu/mol.
In conclusion the paper is of some interest to a broader readership and should thus be published in Materials after improvement of the formal shortcomings.
Author Response
Details of revision and Replies
In our revision, the comments of the three reviewers were accommodated as described below. We thank their comments, which helped improve our work.
[3] Reviewer 3
The paper by Moskin et al. reports on the structural and magnetic properties of Ba2Mn3(SeO3)6. In particular the combination of experimental and theoretical analysis yields a deep insight into the short- and long-range magnetic coupling and spin exchange phenomena. On that basis the properties of this particular material can largely be understood. Furthermore, obtained results will be helpful to understand the properties of structurally similar or closely related materials.
Comment 1:
A minor flow of the presentation is the mixing of cgs and SI units, which could definitely be avoided. Furthermore, the susceptibility=M/H is usually a dimensionless quantity rather than being measured in emu/mol.
Reply 1:
It is generally accepted to use emu/mol to designate quantitively correct measurement of magnetic susceptibility. Using Tesla instead of 104 Oersted is also generally acceptable.
In conclusion the paper is of some interest to a broader readership and should thus be published in Materials after improvement of the formal shortcomings.
Round 2
Reviewer 1 Report
The answers are clear and this is now good for publication.